# DNMT3A Mutation-Induced CDK1 Overexpression Promotes Leukemogenesis by Modulating the Interaction between EZH2 and DNMT3A

**DOI:** 10.3390/biom11060781

**Published:** 2021-05-22

**Authors:** Ying Yang, Yujun Dai, Xuejiao Yang, Songfang Wu, Yueying Wang

**Affiliations:** Shanghai Institute of Hematology, State Key Laboratory of Medical Genomics, National Research Center for Translational Medicine at Shanghai, Rui Jin Hospital, Shanghai Jiao Tong University School of Medicine, Shanghai 200025, China; yangying890612@163.com (Y.Y.); daiyj@sysucc.org.cn (Y.D.); xuejiao_sjtu@163.com (X.Y.); sfwu@shxh-centerlab.com (S.W.)

**Keywords:** acute myeloid leukemia, DNA methyltransferase 3A, mutation, CDK1, targeted therapy

## Abstract

*DNMT3A* mutations are frequently identified in acute myeloid leukemia (AML) and indicate poor prognosis. Previously, we found that the hotspot mutation DNMT3A R882H could upregulate CDK1 and induce AML in conditional knock-in mice. However, the mechanism by which CDK1 is involved in leukemogenesis of DNMT3A mutation-related AML, and whether CDK1 could be a therapeutic target, remains unclear. In this study, using fluorescence resonance energy transfer and immunoprecipitation analysis, we discovered that increased CDK1 could compete with EZH2 to bind to the PHD-like motif of DNMT3A, which may disturb the protein interaction between EZH2 and DNMT3A. Knockdown of CDK1 in OCI-AML3 cells with *DNMT3A* mutation markedly inhibited proliferation and induced apoptosis. CDK1 selective inhibitor CGP74514A (CGP) and the pan-CDK inhibitor flavopiridol (FLA) arrested OCI-AML3 cells in the G2/M phase, and induced cell apoptosis. CGP significantly increased CD163-positive cells. Moreover, the combined application of CDK1 inhibitor and traditional chemotherapy drugs synergistically inhibited proliferation and induced apoptosis of OCI-AML3 cells. In conclusion, this study highlights CDK1 overexpression as a pathogenic factor and a potential therapeutic target for DNMT3A mutation-related AML.

## 1. Introduction

As a member of the DNA methyltransferase family, DNA (cytosine-5)-methyltransferase 3A (DNMT3A), together with DNMT3B, is responsible for de novo DNA methylation. DNMT3A catalyzes the transfer of methyl groups to the CpG-rich regions of active chromatin, leading to gene inactivation, and participates in cell differentiation, development, preservation of chromosomal integrity, parental imprinting, and X-chromosome inactivation [1,2]. Somatic mutations in *DNMT3A* have been identified in approximately 25% of patients with acute myeloid leukemia (AML), with Arg882 (R882) being the hotspot [3,4]. *DNMT3A* mutation occurs in the early stages of AML and is regarded as a pre-leukemic gene mutation [5]. DNMT3A knockout mice showed stronger self-renewal ability of hematopoietic stem cells and significantly enhanced the proliferative ability of myeloid cells [6,7]. DNMT3A R882 mutations (e.g., R882H/C) showed a dominant negative function by disturbing the formation of a tetramer with DNMT3L and inhibiting its own methyltransferase activity [8]. Lu et al. reported that DNMT3A R882 mutation deregulated DNA methylation of crucial AML-promoting genes, such as *Meis1*, *Mn1*, *Hoxa7*, and *Mycn* [9]. In our previous study, we established a bone marrow transplantation mouse model with DNMT3A R882H [10] and a DNMT3A R878H conditional knock-in mouse model [11], which induced chronic myelomonocytic leukemia and AML, respectively. DNMT3A R882H deregulated methylation in the DNA gene body region of *mTOR*, resulting in *mTOR* upregulation. Subsequently, overexpressed mTOR inhibited lysosomal degradation of the CDK1 protein, resulting in CDK1 accumulation. Wei Y et al. showed that CDK1 protein phosphorylated Zeste Homolog 2 (EZH2) at T487, driving EZH2 dissociation from polycomb repressive complex 2 (PRC), and downregulated EZH2 histone methyltransferase activity against tri-methylation of lysine 27 on histone H3 (H3K27me3), which released the inhibition on transcription of the *HOXA* family genes [12]. Studies have shown that *HOXA* family genes play important roles in regulating cell differentiation and are involved in leukemogenesis. Therefore, these data highlight the importance of CDK1 upregulation in the pathogenesis of DNMT3A mutation-related AML. 

Additionally, along with the increase in CDK1 protein, CDK1 was identified in the products immunoprecipitated by DNMT3A antibody [10], which suggested an interaction between DNMT3A protein and CDK1 protein. In the validation test, CDK1 formed a complex with both WT DNMT3A and DNMT3A R882H. DNMT3A contains three functional domains: one catalytic domain in the C-terminal region, namely the SAM-dependent methyltransferase C5-type domain, which recognizes unmethylated DNA and methylates DNA at the C5 position of cytosine, and two in the N-terminal regulatory region, namely a PWWP domain that plays a key role in the binding of DNA and histone, mainly in the binding of tri-methylation of lysine 36 on histone H3 (H3K36me3), as well as a PHD-like ATRX-DNMT3-DNMT3L (ADD) domain that interacts with unmodified histone H3 [13,14]. The ADD domain can also interact with the DNA-binding region of DNMT3A’s catalytic domain showing auto-inhibitory activity, which is released by the binding of the H3 tail peptide to the ADD domain [15]. EZH2 interacts with the conserved PHD-like motifs in DNMT3A and DNMT3B, and direct physical contact between EZH2 and DNMT3A is necessary to control CpG methylation of promoters in the EZH2 target genes *MYT1, WNT1, KCNA1*, and *CRN1* [16]. Whether CDK1 can bind directly to DNMT3A requires further investigation. We propose a hypothesis that there is a competitive relationship between CDK1 and EZH2 in binding DNMT3A, which might influence downstream gene regulation.

AML is a heterogeneous disease with poor survival and a high relapse rate. With advances in understanding the pathogenesis of AML, novel potential therapies for AML, including the refinement of conventional cytotoxic chemotherapies, genetic and epigenetic targeted drugs, and immunotherapies, have been developed in recent years [17,18,19]. However, the treatment of AML with *DNMT3A* mutations still faces challenges. DNMT3A R882H is a loss of function mutation; therefore, to kill DNMT3A R882H AML cells, it might be better to target its downstream-regulated key genes rather than DNMT3A R882H itself. We found that the mTOR–CDK1–EZH2–H3K27me3 axis is involved in the pathogenesis of DNMT3A mutation-related AML [11]. As a member of the cyclin-dependent kinases, CDK1 plays a pivotal role in the cell cycle pathway, driving the cell cycle through the G2 and M phases [20]. The expression of CDK1 was much higher in relapsed AML patients than in newly diagnosed AML patients. Increased nuclear CDK1 correlates with poor outcomes and lower complete remission (CR) rates in AML [21]. Knockdown of CDK1 or inhibition of its activity with the CDK1 selective inhibitor CGP74514A (CGP) and pan-CDK inhibitor increased H3K27me3 and inhibited transcription of the *HOXA* family [12]. CGP can cause cell cycle arrest and promote apoptosis in U937 cells [22]. Flavopiridol (FLA) (also called alvocidib), a pan-CDK inhibitor (CDK1/2/3/4/6/7/9) [23], has shown anti-tumor effects and has been used in clinical trials for some solid tumors [24] and hematological malignancies, including chronic lymphocytic leukemia [25,26], AML [27,28,29], refractory multiple myeloma (clinicalTrials.gov, NCT00047203), and lymphoma (clinicalTrials.gov, NCT00445341). Therefore, we investigated the mechanism of CDK1 overexpression in leukemogenesis and the anti-tumor efficacy of CDK1 inhibitors in DNMT3A mutation-related AML.

## 2. Materials and Methods

### 2.1. Plasmids Construction

CDK1-GFP, EZH2-GFP, and DNMT3A-BFP were constructed by cloning the full-length coding sequences of cDNA of CDK1 and EZH2 into the multiple cloning sites of plasmids pEGFP-C1, and DNMT3A into pEBFP-C1, respectively. To construct truncated and full-length WT-DNMT3A-Flag plasmids, we edited PWWP, ADD, and SAM-dependent MTase C5-type domains by amplifying different lengths of WT DNMT3A sequences and cloning these segments into multiple cloning sites of Flag-CMV6. Flag-tagged fragments consisting of DNMT3A amino acids 1–194, 195–430, 431–610, 611–912, 194–912, 260–912, and 430–912 derived from WT DNMT3A were inserted into eukaryotic expression vector CMV6 for transient transfection assays (Figure 1B). Truncated mutation plasmids of DNMT3A (∆PHD/∆GATA) (Figure 2A) were generated followed by Lightning Site-Directed Mutagenesis Kit (QuikChange^®^ #200518, Stratagene, Unite state and Canada). WT DNMT3A and DNMT3A R882H plasmids were constructed as described in our previous study [10]. All primers used are summarized in Table 1 below.

### 2.2. Cell Culture, Transfection, RNA Interference and Immunoprecipitations

NIH3T3 and 293T cells were cultured in Dulbecco’s modified Eagle’s medium supplemented with 10% fetal bovine serum (FBS) at 37 °C in 5% CO2. Human OCI-AML3 cells were cultured in Ro-swell Park Memorial Institute 1640 medium supplemented with 10% FBS at 37 °C in 5% CO2. NIH3T3 and 293T cells were transfected with different expression plasmids in 10 cm^2^ dishes for 48 h using Lipofectamine 2000 reagent (Invitrogen, Waltham, MA, USA). The cells were lysed with IP lysis buffer (Thermo Scientific Pierce, Rockford, IL, USA). Immunoprecipitations were incubated overnight with anti-CDK1 antibody and anti-EZH2 antibody (Cell Signaling Technology, Boston, MA, USA) with 10 μg of protein-G agarose (Beyotime, Shanghai, China) or anti-FLAG-M2 beads (Sigma-Aldrich, St. Louis, MO, USA). Gels with immunoprecipitated products were washed five times by cold IP lysis buffer; immunoprecipitated proteins were subjected to SDS-PAGE and followed by immunoblot analysis. OCI-AML3 cells were transfected with scrambled small interfering RNA (siRNA) as a negative control (NC) and siRNA targeting CDK1 (siCDK1) for 48 h using Lipofectamine 2000 according to the manufacturer’s instructions. Scrambled siRNA and siCDK1 were purchased from Shanghai GenePharma Co.,Ltd. The oligoribonucleotide sequences of the human siCDK1 were as follows: 5′-GAUGUAGCUUUCUGACAAAAAtt-3′(sense) and 5′-UUUGUCAGAAAGCUACAUCUUtt-3′ (antisense).

### 2.3. Immunofluorescence and FRET Analysis

Monolayer cells of NIH3T3 were fixed with 4% paraformaldehyde and stained with Hoechst for nuclei. To analyze the fluorescence resonance energy transfer (FRET) efficiency, the donor (DNMT3A-BFP) fluorescence was excited at 405 nm, and the acceptor (CDK1-GFP) fluorescence was recorded with a 509 nm cutoff filter. After photobleaching of GFP fluorescence, the BFP fluorescence at 405 nm was recorded again. The gain of energy transfer as CDK1-GFP binds to DNMT3A-BFP was measured as an increase in acceptor fluorescence intensity. The corrected FRET image and FRET efficiency were produced by LAS AF lite(vesion 4.0, Leica Microsystems CMS GmbH, Weztlar, Germany), as reported previously [30]. The kinetic traces shown are averages of 4–6 independent measurements. Next, 293T cells were fixed with 4% paraformaldehyde and incubated with the primary antibodies anti-FLAG and anti-CDK1 (Abcam, Cambridge, England). After washing, cells were incubated with the corresponding fluorescent antibodies as secondary antibodies and stained Hoechst for nuclei. The fluorescence photos were taken with a confocal laser scanning microscope (Leica TCS SP8).

### 2.4. Immunoblotting

Cell lysates were harvested and incubated overnight with the antibodies anti-CDK1, anti-EZH2, anti-FLAG, anti-PARP, anti-Mcl-1, anti-survivin, and anti-actin (Cell Signaling Technology, Boston, USA), respectively. Immunoblotting was performed according to standard protocols as previously described [10]. The chemiluminescence system was used to visualize and quantify the protein bands using ImageJ software.

### 2.5. Flow Cytometric Analysis

Apoptosis was detected using the Annexin V and Propidium Iodide (PI)/7-AAD Apoptosis Detection Kit (BD Biosciences, Franklin Lakes, NJ, USA). Cells were stained with PI (Sigma, St Louis, MO, USA) for cell cycle analysis and CD14-FITC, CD11b-APC, CD86-PE, and CD163-BV421 (BioLegend, San Diego, CA, USA) for differentiation analysis. All the preparation work before flow cytometric analysis was performed according to the corresponding manufacturer’s instructions. Flow cytometric data were collected using an LSR II flow cytometer (Becton Dickinson, Franklin Lakes, NJ, USA) and analyzed using FlowJo software V10 (Becton Dickinson, Franklin Lakes, NJ, USA). 

### 2.6. Chemical Reagents and Drug Combination Analysis

Cells were treated with CGP74514A (Calbiochem, Gibbstown, NJ, USA) and flavopiridol (Selleck, Houston, TX, USA) dissolved in DMSO (Sigma-Aldrich, St. Louis, MO, USA), and cytarabine (Ara-C, Pfizer, NY, USA), doxorubicin (Dox, Pfizer, NY, USA), and homoharringtonine (HHT, MINGSHEN PHARMA, Hangzhou, China) dissolved in 0.9% NaCl at various concentrations in a 96-well plate. In all experiments, the final concentration of DMSO did not exceed 0.1%. For drug combination studies, OCI-AML3 cells were treated with the combination of two agents at a fixed ratio for 48 h. Inhibition rate of cell proliferation was evaluated by the Cell Counting Kit-8 (CCK8/WST-8) kit (Biotool, Hamburg, Germany), according to the manufacturer’s instructions. The concentrations at which the agents inhibited cell viability by 50% (IC50) were determined by using the CompuSyn software (Biosoft, Ferguson, MO, USA). The drug combination experiments were conducted and the combination index (CI) was calculated by using the CompuSyn software as previously described [31]. Regarding CI data interpretation, CI < 1.0 indicates synergism, CI = 1.0 means addition, and CI > 1.0 means antagonism. Fraction affected (Fa) means the inhibition rate of cell viability.

### 2.7. Statistical Analysis

The statistical significance of differences between the means of two groups was evaluated by the two-tailed t-test. Multiple comparison was analyzed by using the one-way analysis of variance (ANOVA) with a post-hoc test. All statistical analyses were performed using GraphPad Prism software (version 7.0; GraphPad Software, Inc., La Jolla, CA, USA).

## 3. Results

### 3.1. DNMT3A Protein Bound to CDK1 Protein through the ADD Domain

To explore the interaction between DNMT3A and CDK1 proteins in situ, we established cell models expressing both DNMT3A-BFP and CDK1-GFP proteins. In the NIH3T3 cell model, confocal fluorescence microscopy showed that the green fluorescence and blue fluorescence overlapped, suggesting the colocalization of DNMT3A and CDK1 proteins. Meanwhile, we observed that the two fluorescent protein tags formed a pair for fluorescence resonance energy transfer (FRET), and the FRET efficiencies of cells transfected with DNMT3A-BFP and CDK1-GFP were obviously higher than those of cells transfected with DNMT3A-BFP and vector-GFP constructs (8.22 ± 5.57% vs. 29.46 ± 9.17%) (Figure 1A), which confirmed the direct interaction between DNMT3A and CDK1 in situ. To map the domain of DNMT3A responsible for the binding of CDK1, we performed deletion mutant analysis. The DNMT3A protein consists of three conserved domains: the PWWP domain and the ADD domain at the N-terminal regulatory region, and the catalytic domain at the C-terminal region. Then, we constructed truncated DNMT3A plasmids with flags by editing the PWWP domain, ADD domain, and SAM-dependent MTase C5-type domain to map the DNMT3A domain responsible for binding to CDK1 (Figure 1B). Since EZH2 protein has been verified to directly bind to DNMT3A and DNMT3B proteins [16], we used EZH2 as a positive control. Co-immunoprecipitation revealed that in 293T cells, the truncated DNMT3A without the ADD domain (including plasmids 4, 5, and 6 as indicated) can hardly precipitate CDK1 protein or EZH2 protein (Figure 1C). Thus, CDK1 protein can bind to DNMT3A protein directly depending on the ADD domain’s similarity to EZH2 protein.

**Figure 1 biomolecules-11-00781-f001:**
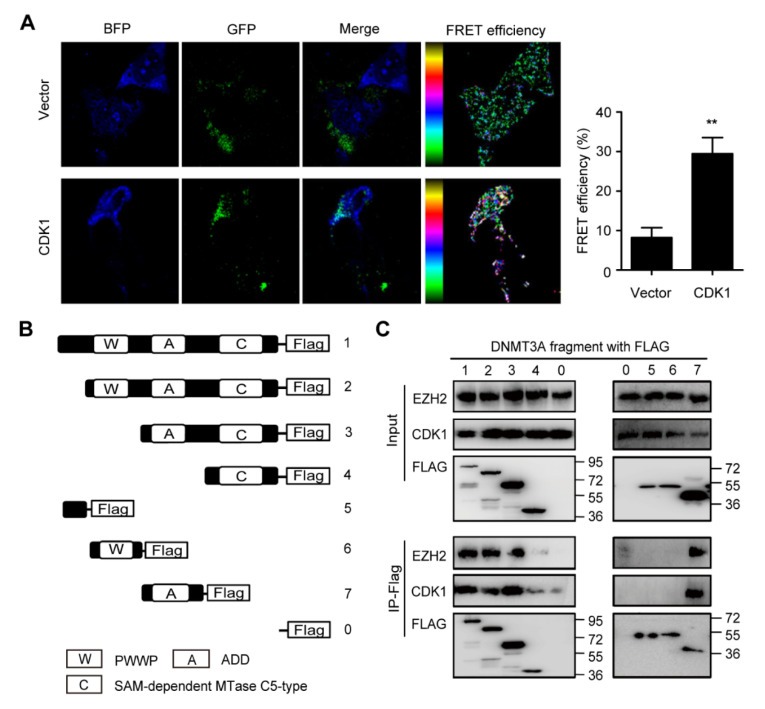
ADD domain of DNMT3A is responsible for the interaction between DNMT3A and CDK1. (**A**) NIH3T3 cells expressed Vector-GFP, CDK1-GFP (green), and DNMT3A-BFP (blue) as indicated. The merged picture reveals the colocalization of CDK1 and DNMT3A proteins. The fluorescence resonance energy transfer (FRET) analysis demonstrated that the FRET efficiency increased more in the CDK1 group than in the vector group. (**B**) Truncated DNMT3A plasmids were constructed by editing PWWP, ADD, and SAM-dependent MTase C5-type domains as indicated. (**C**) Co-immunoprecipitation of CDK1, EZH2, and truncated DNMT3A/wild-type DNMT3A proteins as indicated. Data are presented as mean ± SD, ** *p* < 0.01.

### 3.2. DNMT3A R882H Mutation-Induced CDK1 Overexpression Interrupted the Binding of EZH2 to PHD Domain of DNMT3A

The ADD domain of DNMT3A consists of two zinc finger motifs, PHD and GATA. EZH2 has been reported to interact with the conserved PHD-like motif in DNMT3A and DNMT3B [16]. To further identify the exact motif of DNMT3A interacting with CDK1, we constructed truncated DNMT3A plasmids: ∆PHD (PHD deleted) and ∆GATA (GATA deleted) (Figure 2A). Immunofluorescence staining revealed that CDK1 (green fluorescence), WT DNMT3A, and ∆GATA DNMT3A (red fluorescence) were expressed in the cytoplasm and nucleus of 293T cells, while ∆PHD DNMT3A was located in the nucleus. CDK1 was mainly co-localized with WT DNMT3A and ∆GATA DNMT3A in the cytoplasm (yellow fluorescence) and hardly in the nucleus. No co-localization was observed between ∆PHD DNMT3A and CDK1 (Figure 2B). Co-immunoprecipitation showed that ∆PHD could not precipitate either CDK1 or EZH2 protein, unlike WT DNMT3A and ∆GATA (Figure 2C). This suggested that once the PHD zinc finger was deleted, the co-localization of CDK1 and DNMT3A was disrupted and DNMT3A’s cytoplasmic localization might be related to its interaction with CDK1. Thus, PHD is the key structure of DNMT3A, which interacts with CDK1. As both CDK1 and EZH2 interact with DNMT3A through the PHD zinc finger, we wondered whether CDK1 could compete with EZH2 to interact with DNMT3A and designed an assay by expressing FLAG-tagged DNMT3A and GFP-tagged CDK1 in 293T cells. We found that when GFP-tagged CDK1 was increased, FLAG-tagged DNMT3A precipitated more CDK1 but less EZH2 protein (Figure 2D). What happens when DNMT3A is mutated? We expressed WT DNMT3A or DNMT3A R882H mutant in 293T cells; a co-immunoprecipitation experiment showed that DNMT3A R882H mutant bound more CDK1 and less EZH2 than WT DNMT3A (Figure 2E). This suggests that the DNMT3A R882H mutation contributes to CDK1 protein accumulation and that excess CDK1 interrupts the binding of EZH2 to DNMT3A.

**Figure 2 biomolecules-11-00781-f002:**
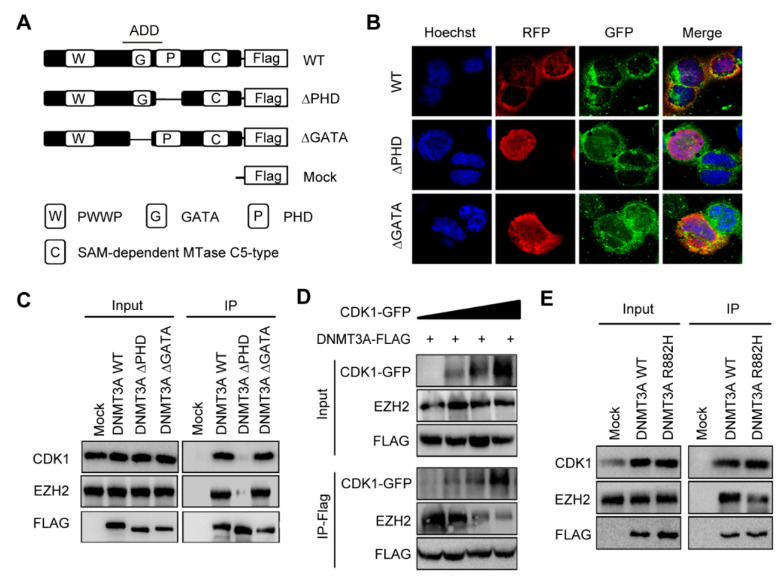
CDK1 overexpression interrupted the binding of EZH2 to PHD domain of DNMT3A. (**A**) Truncated DNMT3A-∆PHD and ∆GATA were constructed by editing ADD domain as indicated. (**B**) Immunofluorescent localization of CDK1, WT DNMT3A, ∆GATA, and ∆PHD proteins in 293T cells. (**C**) Co-immunoprecipitation of CDK1, EZH2, and DNMT3A proteins in 293T cells expressing truncated DNMT3A or WT DNMT3A. (**D**) Co-immunoprecipitation of CDK1, EZH2, and wild-type DNMT3A proteins in 293T cells transfected with FLAG-tagged DNMT3A(DNMT3A-FLAG) and GFP-tagged CDK1 (CDK1-GFP) as indicated. (**E**) Co-immunoprecipitation of CDK1, EZH2, and DNMT3A proteins in 293T cells expressing DNMT3A R882H or WT DNMT3A.

### 3.3. CDK1 Inhibitor Could Arrest Cells in G2M Phase and Induce Apoptosis in DNMT3A Mutation AML Cells

Here, we conclude an mTOR–CDK1–EZH2–H3K27me3 regulation axis in AML (Figure 3). DNMT3A mutation led to decreased DNA methylation of *mTOR*, resulting in overexpression of mTOR. The overexpressed mTOR inhibits the degradation of CDK1 protein by lysosomes, causing CDK1 protein accumulation. First, as a cell cycle protein, CDK1 promotes cell division and proliferation. Second, increased CDK1 phosphorylated EZH2 and inhibited EZH2 methylation on H3K27, which released the H3K27me3 depression on the HoxA family, and the expression of the HoxA family impeded cell differentiation. Third, the excess CDK1 protein competes with EZH2 protein in binding to the DNMT3A protein. Additionally, it has been verified that EZH2 can physically interact with DNMT3A, recruit DNMT3A to the promoters of its target genes, and facilitate CpG methylation and regulate the expression of MYT1, WNT1, KCNA1, and CNR1 [16]. The interruption of EZH2 binding to DNMT3A may impede EZH2 methylation of DNMT3A on the promoter of WNT1, leading to overexpression of WNT1. WNT1 overexpression can activate the WNT pathway and lead to the transcription of its downstream genes, *c-myc*, promoting tumor cell division and migration [32]. Thus, DNMT3A mutation can interact directly and indirectly with CDK1 to promote CDK1 function and restrain EZH2 function, which suggests that CDK1 can promote DNMT3A mutation-related AML progression in multiple ways.

To further investigate the function of CDK1 in DNMT3A mutation-related AML, we knocked down gene expression levels of CDK1 using siRNA against OCI-AML3 cells. The CDK1 protein expression was significantly decreased by siCDK1 (Figure 4A). Meanwhile, CCK8 analysis showed that cell proliferation was decreased (Figure 4B) and Annexin V/7-AAD staining showed that apoptosis was increased (Figure 4C) in the siCDK1 group compared with the NC group, suggesting that CDK1 knockdown can hinder *DNMT3A* mutation-related AML progression. Then, we examined the effect of selective CDK1 inhibitor CGP and pan-CDK inhibitor CDK1 on the biological characteristics of the OCI-AML3 cells by flow cytometry. The IC50 of CGP and FLA in DNMT3A-mutated cell line OCI-AML3 was 1.03 µM and 36.82 nM, respectively. We treated OCI-AML3 cells with 0.8 and 1.2 µM CGP or 30 and 60 nM FLA and performed cell cycle analysis at 4, 8, 12, 18, and 24 h. Increased cells fraction in the G2/M phase was observed at 4 h, reaching a peak in the 1.2 µM CGP group by 4 h, and in the 60 nM FLA group by 18 h (Figure 5A). Twenty-four hours after treatment with 0.8 and 1.2 µM CGP or 30 and 60 nM FLA, Annexin V and PI staining showed that the apoptosis of OCI-AML3 cells significantly increased (Figure 5B). To analyze the differentiation of OCI-AML3 cells, CD14, CD11b, CD86, and CD163 expressions on the cell surface were detected after treatment of OCI-AML3 cells with CGP or FLA at 24, 48, and 72 h, respectively. The expression levels of CD14, CD11b, and CD86 were not obviously changed, while CD163 expression was significantly increased, 48 h after treatment with CGP, but not FLA (Figure 5C). Giemsa staining of OCI-AML3 cells revealed no obvious change in morphology. Thus, CDK1 inhibitors can arrest cell cycle and induce apoptosis in AML cells with DNMT3A mutation.

### 3.4. CDK1 Inhibitor and HHT Synergically Inhibited Proliferation of OCI-AML3 Cells

To investigate whether CDK1 can be combined with traditional chemotherapy drugs to treat DNMT3A related AML, we conducted pharmaceutical experiments by combining CGP or FLA with the chemotherapy drugs cytarabine (Ara-C), doxorubicin (Dox), or homoharringtonine (HHT). The IC50 of Ara-C, Dox, and HHT in OCI-AML3 cells was 41.60 ng/mL, 1.73 µM, and 16.27 nM, respectively. Cells were treated with single agent, or both agents at 4, 2, 1, 0.5, and 0.25 times the IC50. The drug combination causing 25, 50, 75, and 90% inhibition of proliferation indicates Fa 0.25, Fa 0.50, Fa 0.75, and Fa 0.90, respectively. For OCI-AML3 cells, combinations of CGP + Ara-C, CGP + Dox, CGP + HHT, FLA + Ara-C, FLA + Dox, and FLA + HHT all exerted synergistic effects at 75–90% inhibition levels, whereas CGP + Ara-C and FLA + Ara-C at 50% inhibition level, and CGP + Ara-C, CGP + Dox, FLA + Ara-C, and FLA + Dox at 25% inhibition level showed antagonistic effects, as indicated by their CI values (Figure 6A). In particular, we observed that all the CI values of CGP or FLA combined with HHT were less than 1.0 and decreased as Fa increased, indicating that both CGP and FLA show strong synergism with HHT. Thus, it seems that the CDK1 inhibitor combined with HHT is a better strategy for combination therapy than Ara-C and Dox. Additionally, we conducted apoptosis analysis using CDK1 inhibitors combined with HHT. Apoptosis of OCI-AML3 cells dramatically increased (Figure 6B) and the cleaved Parp protein significantly increased (Figure 6C) after 24 h of exposure to the combination of CDK inhibitor and HHT, compared with the DMSO and single agent groups. Furthermore, we observed that survivin and MCL-1 protein levels increased after treatment with CGP or FLA, which was reversed when combined with HHT (Figure 6C). This may explain why the combination of CGP or FLA and HHT showed significant synergism compared to the other drugs.

## 4. Discussion

In this study, we propose that CDK1 can bind to DNMT3A directly through the PHD domain. The PHD-like domain is conserved in all three DNMTs (DNMT3A, DNMT3B, and DNMT3L) and has been previously demonstrated to be instrumental for the recruitment of histone methyltransferase to the unmethylated H3K4, which are present on promoters of bivalent genes [33]. As a histone-lysine N-methyltransferase enzyme, EZH2 can also interact with the conserved PHD domain in DNMT3A and DNMT3B and work together to regulate target genes [16]. We further revealed that CDK1 and EZH2 competitively bind to DNMT3A, and accumulated CDK1 disrupted EZH2 binding to DNMT3A. EZH2-depletion contributes to significantly reduced methylated CpGs located within WNT1 [21]. We speculate that the constitutively activated WNT/β-catenin pathway in AML [34] might result from WNT1 overexpression by EZH2 loss of function, as WNT1 can activate the Wnt pathway [32]. Thus, in addition to its direct phosphorylation on EZH2, we propose a new method by which CDK1 proteins can directly grab more DNMT3A proteins and disturb the cooperation between EZH2 and DNMT3A after DNMT3A mutation. Activation of the Wnt pathway may play a role in DNMT3A mutation-related AML.

DNMT3A-mutated AML patients have shorter overall survival than non-DNMT3A-mutated patients and show an unfavorable prognosis [35]. DNMT3A mutant transcript levels persist in chemotherapy-induced remission in AML [36], which indicates that DNMT3A mutation clones are not easy to clear. In recent years, a variety of epigenetic target therapy drugs have entered clinical trials, such as isocitrate dehydrogenase (IDH) inhibitors AG120 and AG-221, histone deacetylase (HDAC) inhibitor, DNA methyltransferase inhibitors azacitidine and decitabine, etc. [37]. There is no drug targeting DNMT3A mutation yet; it is difficult to target DNMT3A mutation directly because DNMT3A R882 mutation is a loss-of-function mutation. Alternatively, researchers have chosen to target downstream genes. Rau et al. treated primary cells from DNMT3A mutation-related AML patients with H3K79 methyltransferase Dot1L inhibitor EPZ5676; Dot1L is the key downstream gene overexpressed in hematopoietic stem cells, which regulates leukemia-related genes and promoted the development of disease in their Dnmt3a knockout murine, which significantly inhibited colony formation and induced terminal differentiation of the primary cells [38]. In our previous work, we targeted the key downstream regulatory gene *mTOR* of DNMT3A mutation with the mTOR inhibitor rapamycin, which significantly inhibited the proliferation of DNMT3A-mutated AML cell lines and prolonged the survival of Dnmt3a R878H AML mice [11]. Here, we targeted CDK1 with selective CDK1 inhibitor CGP and pan-CDK inhibitor FLA to recover EZH2 function, which significantly arrested cells in G2/M phase, and promoted apoptosis of DNMT3A-mutated AML cells. Targeting CDK1 has been reported to promote FLT3-activated acute myeloid leukemia differentiation in cell lines as well as in patient blood samples [39]. However, in our study, the classical myeloid differentiation markers, CD14 and CD11b, remained unchanged. Although CD163 was slightly increased, no macrophage cell morphology changes were observed in Giemsa staining analysis.

Although Ara-c and anthracycline antibiotic-based combination chemotherapy can induce complete remission (CR) in 40–80% of patients with previously untreated AML, the majority of older patients could not benefit from it due to poor tolerance and high mortality [40]. Thus, many combined therapies have been proposed to decrease the dose of traditional chemotherapy drugs. Loss of EZH2 has been reported to induce resistance to multiple drugs in AML [41]. Our drug combination experiment showed that the CDK1 inhibitor exerted a synergistic effect with the traditional chemotherapy drugs Ara-C, Dox, and HHT. HHT was reported to induce the decreased synthesis of short-life proteins that are related to proliferation and apoptosis (e.g., c-Myc, Mcl-1, cyclin D1, β-catenin, Bcl-Abl, etc.), and has been used in clinical trials of AML for decades [42]. HHT-based drug combinations such as HA (HHT and Ara-C) and HAA (HHT, Ara-c, and aclarubicin) showed effective results and low toxicity in young adults and elderly patients with AML [43,44]. We found that HHT disrupted the expression of MCL-1 and suvivin proteins induced by CGP and FLA. Thus, CDK1 inhibition combined with HHT may be suggested in DNMT3A mutation-related AML. Further in vivo experiments are needed to validate our results. The CDK4/6 inhibitors palbociclib, ribociclib, and abemaciclib have been recently approved by the Food and Drug Administration (FDA) for the treatment of breast cancer [45]. A naturally derived small molecule CDK1 and AKT inhibitor terameprocol has shown safety and a partial response in some advanced leukemia patients [46]. Therefore, we would like to suggest in vivo verification and further possible preclinical and clinical studies of CDK1 inhibitors be carried out in DNMT3A mutation-related AML.

## 5. Conclusions

Collectively, DNMT3A mutations can regulate gene expression not only by directly regulating the DNA methylation level, but also by indirectly affecting histone modification-marked epigenetic regulation, and CDK1 may be the core of the regulation axis. Targeting CDK1 can hinder DNMT3A mutation-related AML progression and may be a promising approach in DNMT3A mutation-related AML therapy.

## Figures and Tables

**Figure 3 biomolecules-11-00781-f003:**
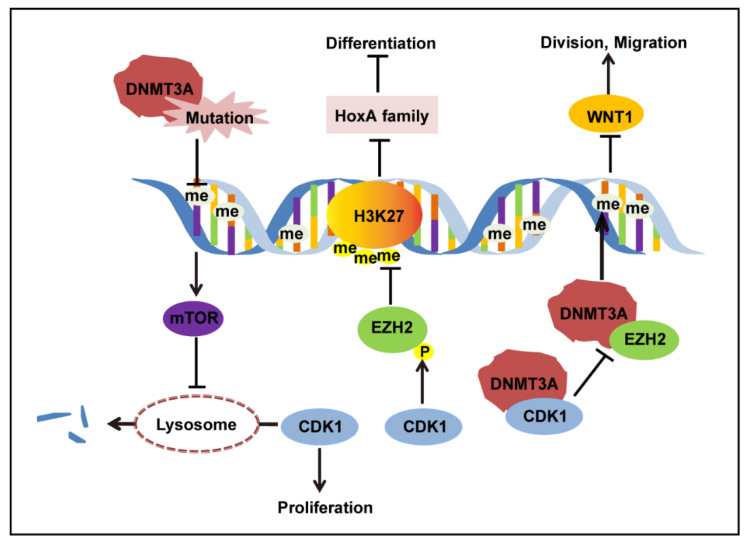
CDK1 regulation in the context of DNMT3A R882 mutation.

**Figure 4 biomolecules-11-00781-f004:**
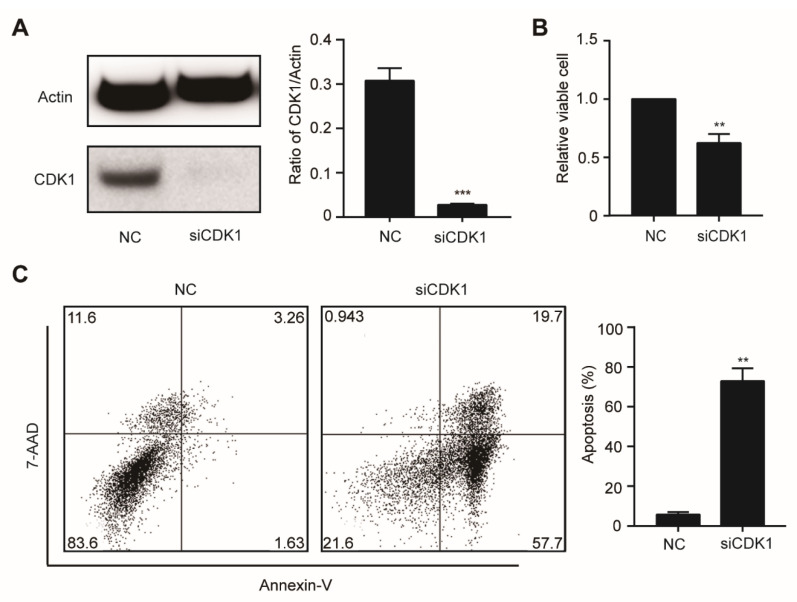
The knockdown of CDK1 inhibits proliferation and induces apoptosis in OCI-AML3 cells. (**A**) The OCI-AML3 cells were transfected with scrambled siRNA (negative control, NC) and siCDK1, respectively, and examined by Western blotting using the indicated antibodies. (**B**) Cell proliferation was analyzed by the CCK-8 assay. (**C**) Apoptosis rate was assessed by Annexin V/7-AAD staining and flow cytometry. Data are presented as mean ± SD, ** *p* < 0.01, *** *p* < 0.001.

**Figure 5 biomolecules-11-00781-f005:**
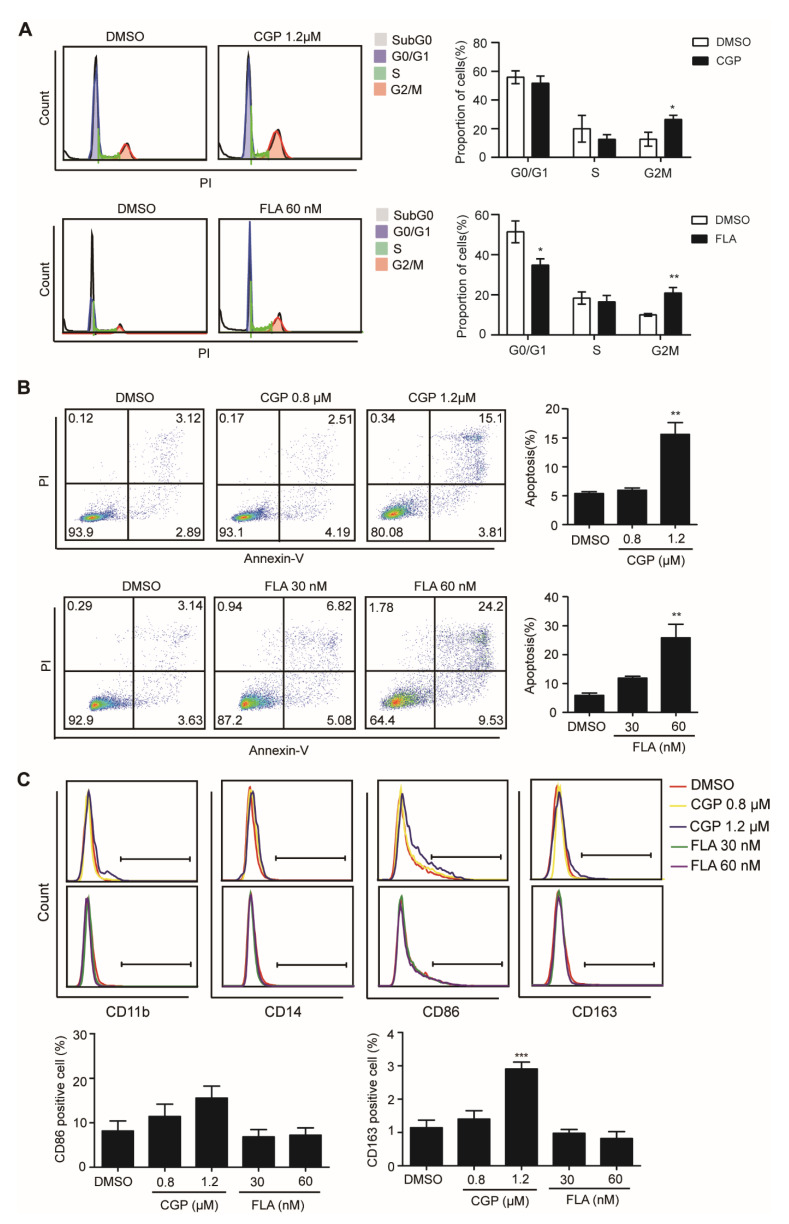
CDK1 inhibitors CGP and FLA can cause cell cycle arrest, apoptosis, and differentiation markers’ expression of OCI-AML3 cells. Flow cytometric analysis on cell cycle (**A**), apoptosis (**B**), and differentiation (**C**) of OCI-AML3 cells after treatment with CGP, FLA, or DMSO as indicated. The cell cycle analysis was conducted after treatment with CGP and FLA for 4 and 18 h, respectively. The apoptosis analysis was conducted after treatment with CGP or FLA for 24 h. CD11b, CD14, CD86, and CD163 were detected after treatment with CGP or FLA for 48 h. Data are presented as mean ± SD, * *p* < 0.05, ** *p* < 0.01, *** *p* < 0.001.

**Figure 6 biomolecules-11-00781-f006:**
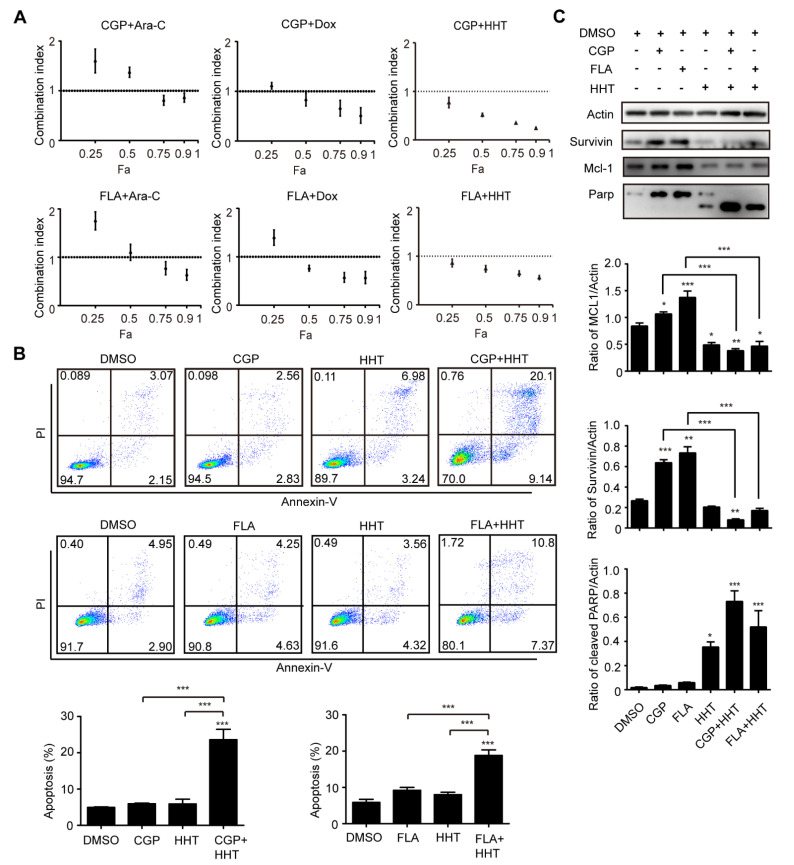
CGP and FLA show strong synergy with HHT in OA3 cells. (**A**) Combination Index plots obtained from OCI-AML3 cells exposed simultaneously to chemotherapy drugs (Ara-C, Dox, or HHT) and CDK1 inhibitors (CGP or FLA) for 24 h as indicated. (**B**) The apoptosis analysis of OA3 with different treatments of CGP, FLA, and HHT for 24 h as indicated. (**C**) Western blotting analysis for the apoptosis-related protein (Parp, MCL-1, and survivin) in OA3 cells treated with CGP, FLA, and HHT for 24 h as indicated. Data are presented as mean ± SD, * *p* < 0.05, ** *p* < 0.01, *** *p* < 0.001.

**Table 1 biomolecules-11-00781-t001:** Primers are listed in table below.

Primer Name	Forward	Reverse
EZH2-PEGFP-C1	CCCTCGAGACATGGGCCAGACTGGGAA	CGGGATCCTCAAGGGATTTCCATTTCTCT
CDK1-PEGFP-C1	CGGAATTCGATGGAAGATTATACCAAAATAGAGA	CGGGATCCCTACATCTTCTTAATCTGATTGTCC
WT-DNMT3A-Flag(1)	CGGAATTCCATGCCCGCCATGCCCTCCAGCGGCCCC	GAAGATCTTTACACACACGCAAAATACTCCTTCAGC
195–912-DNMT3A-Flag(2)	CGGAATTCCATGCCCTACTACATCAGCAAGCGCAA	GAAGATCTTTACACACACGCAAAATACTCCTTCAGC
431–912-DNMT3A-Flag(3)	CGGAATTCCATGAATCCCTACAAAGAAGTGTACACGG	GAAGATCTTTACACACACGCAAAATACTCCTTCAGC
611–912-DNMT3A-Flag(4)	CGGAATTCCATGGCTAATAACCACGACCAGGAATTTG	GAAGATCTTTACACACACGCAAAATACTCCTTCAGC
1–194-DNMT3A-Flag(5)	CGGAATTCCATGCCCGCCATGCCCTCCAGCGGCCCC	GAAGATCTTTAGTCCCCCGCCTGGAAGGTGAGCCTCG
195–430-DNMT3A-Flag(6)	CGGAATTCCATGCCCTACTACATCAGCAAGCGCAA	GAAGATCTTTACTTCTCTTCTTCTGGTGGCT
431–610-DNMT3A-Flag(7)	CGGAATTCCATGAATCCCTACAAAGAAGTGTACACGG	GAAGATCTTTAGAAGAACATCTGGAGCCGGGAGGG
∆PHD-DNMT3A-Flag	CGACGACGGCTACACCTACGGGCTGC	GCAGCCCGTAGGTGTAGCCGTCGTCG
∆GATA-DNMT3A-Flag	CCGGAACATTGAGGACTGTGCGTACCAGTACG	CGTACTGGTACGCACAGTCCTCAATGTTCCGG

## Data Availability

The data presented in this study are available on request from the corresponding author.

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
