# Peer review of "DNMT3A Mutation-Induced CDK1 Overexpression Promotes Leukemogenesis by Modulating the Interaction between EZH2 and DNMT3A"

_biomolecules, 2021, doi:10.3390/biom11060781_

Round 1

Reviewer 1 Report

The ms could be acceptable, but it requires only few modifications. I suggest to better describe combination index, in particualr with a different graphical solutions. I also suggest to put more emphasis on the possible applications of the observations.

Author Response

Point 1: I suggest to better describe combination index, in particualr with a different graphical solutions. 

Response 1: We thank the reviewer for the suggestion. We have made corresponding revisions to better describe combination index, as highlighted in line 175-184 on page 5 (the material and methods section) and in line 329-338 on page 11 (the results section).

Point 2: I also suggest to put more emphasis on the possible applications of the observations.

Response 2: We thank the reviewer for the nice suggestion. Our study highlights CDK1 overexpression as a pathogenic factor and a potential-therapeutic target for DNMT3A mutation-related AML. Several CDK inhibitors have been approval for clinical use in recent years. For example, CDK4/6 inhibitors Palbociclib, Ribociclib, and Abemaciclib have been approved by the FDA for the treatment of breast cancer. A naturally derived small molecule CDK1 and AKT inhibitor Terameprocol has showed safety, partial response in some advanced leukemia patients. However, the pharmacological effects of CDK1 inhibitors on DNMT3A mutation-related AML still need to be verified in vivo. We would like to further address this issue in the future as a separate work. In this study, we focused on revealing the molecular mechanisms underlying DNMT3A mutation-induced CDK1 overexpression in leukemogenesis. On the basis of in vivo experiments, we would like to suggest that further possible preclinical and clinical studies of CDK1 inhibitors be carried out in AML patients with DNMT3A mutations. Therefore, we have added some discussion of the possible applications of CDK inhibitors in the discussion section, as highlighted in line 414-419 on page 13.

Reviewer 2 Report

This manuscript entitled “DNMT3A mutation-induced CDK1 overexpression promotes leukemogenesis by modulating the interaction between EZH2 and DNMT3A” described interesting findings and the results can support the conclusions. To strengthen this manuscript, the authors should clarify the following points.
Major comments
1. What is the effect of CDK1-knockdown in DNMT3A-mutated cells on cell growth, apoptosis and differentiation in leukemia cells?
2. Authors could explore the binding effects between DNMT3A (WT and R882H mutant) and CDK1 or EZH2 by molecular modeling analysis to support the data for protein interaction.
3. In this manuscript, the two-tailed t-test is not suitable for Statistic analysis of all data. The one-way or two-way ANOVA with post hoc test for multiple comparison is suggested.
Minor:
In line 268 (page 8), is the IC50 of FLA (36.82 μM) in OCI-AML3 correct?

Author Response

Point 1: What is the effect of CDK1-knockdown in DNMT3A-mutated cells on cell growth, apoptosis and differentiation in leukemia cells?

Response 1:We thank the reviewer for this nice question. We investigated the effects of CDK1-knockdown in OCI-AML3 cells on cell proliferation and apoptosis, and the data have been added in the revised manuscript. Corresponding changes have been highlighted in abstract section (in line 18-19 on page1), materials and methods section (in line 118 on page3  and 130-136 on page 4), results section (in line 283-289 on page 8), and Figure 4 (on page 9).

Point 2: Authors could explore the binding effects between DNMT3A (WT and R882H mutant) and CDK1 or EZH2 by molecular modeling analysis to support the data for protein interaction.

Response 2: We thank the reviewer for this constructive comment. To explore the binding effects between DNMT3A (WT and R882H mutant) and CDK1 or EZH2 by molecular modeling, we need to obtain information on the three-dimensional crystal structures of these proteins. We searched the related literatures and the data from RCSB database. However, we found that there are no three-dimensional crystal structures of the full length DNMT3A (WT and R882H mutant) with CDK1 or EZH2. Investigation on the crystal structures of these protein complexes is necessary for molecular modeling analysis. In addition, biolayer interferometry technique can be used to determine the kinetic parameters of the interaction between DNMT3A and CDK1 or EZH2. The structural and biochemical analyses of binding effects between DNMT3A (WT and R882H mutant) and CDK1 or EZH2 might not only support our data but also provide a molecular basis for drug screening in DNMT3A mutation-related AML. We would like to further address this issue in the future as a separate work.

Point 3: In this manuscript, the two-tailed t-test is not suitable for Statistic analysis of all data. The one-way or two-way ANOVA with post hoc test for multiple comparison is suggested.

Response 3: We thank the reviewer and agree with the comment. According to the reviewer’s suggestion, we performed multiple comparison analysis by using the one-way ANOVA with post hoc test. The updated P values were shown in Fig5C, Fig6B and Fig6C. Corresponding changes have been made in abstract section (in line 21 on page 1), materials and methods section (in line 187-188 on page 5), results section (in line 300-304 on page 8-9), and discussion section (in line 397-398 on page 13), which were highlighted in the revised manuscript.

Minor:In line 268 (page 8), is the IC50 of FLA (36.82 μM) in OCI-AML3 correct?

Response: We apologize for this mistake and are very grateful to the reviewer for the careful review. The IC50 of FLA is 36.82 nM, and the correction has been made in line 292 on page 8 (highlighted).

Round 2

Reviewer 2 Report

Authors have satisfactorily addressed my concerns.